# MiRNA Profiling in Premalignant Lesions and Early Glottic Cancer

**DOI:** 10.3390/cancers17172883

**Published:** 2025-09-02

**Authors:** Anna Rzepakowska, Agnieszka Zajkowska, Marta Mękarska, Julia Śladowska, Aleksandra Borowy, Maciej Małecki

**Affiliations:** 1Otorhinolaryngology Department Head and Neck Surgery, Medical University of Warsaw, 02-097 Warszawa, Poland; 2Department of Applied Pharmacy, Faculty of Pharmacy, Medical University of Warsaw, 02-091 Warszawa, Poland; 3Students’ Scientific Research Group at the Otorhinolaryngology Department Head and Neck Surgery, Medical University of Warsaw, 02-097 Warszawa, Poland; 4Laboratory of Gene Therapy, Faculty of Pharmacy, Medical University of Warsaw, 02-091 Warszawa, Poland

**Keywords:** miRNA, laryngeal dysplasia, laryngeal cancer, prognosis, carcinogenesis

## Abstract

This study delineates microRNA (miRNA) expression dynamics during the progression of laryngeal squamous cell carcinoma (LSCC) from non-dysplastic epithelium to invasive carcinoma. Comprehensive profiling using a broad-spectrum microarray platform revealed distinct, stage-specific signatures. Advanced dysplastic and invasive lesions demonstrated significant upregulation of miR-375, miR-143-3p, miR-145-5p, and miR-26a-5p, whereas miR-126-3p exhibited progressive downregulation. The tumor suppressor candidates miR-216a-5p and miR-488-3p were confined to normal tissue, suggesting their early loss during malignant transformation. Conversely, miR-105-5p and miR-516a-5p were selectively expressed in high-grade lesions and carcinoma. These findings underscore the critical role of miRNAs in LSCC pathogenesis and support their potential utility as biomarkers for early diagnosis, disease monitoring, and therapeutic stratification. Validation in larger patient cohorts and assessment of circulating miRNAs in biofluids are warranted to confirm their clinical applicability in non-invasive screening approaches.

## 1. Introduction

Laryngeal squamous cell carcinoma (LSCC) accounts for nearly one-third of head and neck squamous cell carcinomas (HNSCCs). It is frequently diagnosed at advanced stages (Stage III or IV), where treatment options are limited to radical surgery or chemoradiotherapy. These treatments frequently compromise vocal, breathing, and swallowing functions, which in turn cause substantial physical and psychological burdens and markedly reduce patients’ quality of life. Despite advances in oncology, the five-year survival rate for laryngeal cancer has shown little improvement over the past thirty years. Globally, over 189,000 cases of laryngeal cancer were reported in 2022, with a mortality rate of 103,359 cases [1]. Due to demographic changes, the incidence is projected to increase by more than 50% by 2045 [2].

To reduce the foreseeable morbidity and mortality, it is essential to raise awareness about risk factors, implement screening for at-risk populations, detect early-stage lesions in the carcinogenic process, and ensure effective treatment that preserves organ function. LSCC is the most common malignancy of the upper respiratory tract, with prolonged exposure to tobacco smoke, especially when combined with alcohol abuse, being the main risk factor [3]. Most invasive cancers arise from precancerous dysplastic changes in the laryngeal epithelium. Chronic irritation of the laryngeal mucosa leads to epithelial changes such as inflammation, hypertrophy, and hyperkeratosis with leukoplakia formation, or more rarely, erythroplakia and ulceration. These lesions are typically visualized and diagnosed clinically. Histopathological analysis of laryngeal biopsy specimens confirms progressive epithelial thickening and changes in cell proliferation and differentiation across clinical stages [4]. Based on the proportion of atypical cells in the epithelial layer, the World Health Organization recommends either a two-tiered classification system with low-grade and high-grade dysplasia or a three-tiered system, adding the carcinoma in situ (CIS) stage [4]. Identification of basement membrane invasion signifies early cancer. However, the clinical staging of epithelial changes does not directly correlate with histological grading. Current research seeks reliable molecular diagnostic markers to overcome the subjectivity of clinical evaluations and controversies in biopsy interpretation and to guide appropriate treatment decisions. Undoubtedly, these morphological and cellular changes arise from complex, multistep genetic, epigenetic, and immuno-oncological alterations that are still not fully understood. Additionally, there are no effective methods to identify lesions with a high risk of malignant transformation. Recent reviews have shown that the risk of malignant transformation varies widely across different dysplasia stages: 0–41.7% for low-grade dysplasia, 14.3–44.4% for high-grade dysplasia, and 11.1–75% for CIS [5].

MicroRNAs (miRNAs) are endogenous, non-coding RNA molecules of approximately 18–24 nucleotides in length. They function as key post-transcriptional regulators of gene expression, predominantly through sequence-specific binding to complementary sites within the 3′ untranslated regions (3′-UTRs) of target mRNAs. This binding event typically induces translational inhibition or promotes mRNA destabilization and subsequent degradation [6,7,8]. In many cancers, miRNAs play crucial roles by promoting or suppressing oncogenes or tumor suppressor genes. They influence fundamental biological processes such as cell proliferation, differentiation, and intercellular communication, and their effects can extend to both neighboring and distant tissues. Dysregulation of miRNA expression has been implicated in cancer progression [6,7,8,9].

Thus far, the contribution of miRNAs to the complex, multistep development of laryngeal carcinogenesis has been only partially elucidated [10,11,12]. In this study, we sought to characterize miRNA expression patterns across different stages of laryngeal lesions, including normal controls, with the objective of identifying aberrantly regulated molecules. These may serve as a basis for exploring associated gene pathways and could provide tools for differential diagnosis or act as prognostic markers in early mucosal lesions of the larynx.

We hypothesized that distinct histopathological stages of laryngeal lesions are characterized by specific miRNA expression profiles. To the best of our knowledge, no prior study has systematically explored miRNA profiling in precancerous laryngeal lesions.

Recent advances in miRNA detection technologies have greatly improved the sensitivity, specificity, and speed of profiling these critical regulatory molecules. Traditional methods, such as qRT-PCR and microarrays, remain widely available and relatively affordable, but they can struggle with low-abundance miRNAs and multiplexed analysis. Innovative approaches, including isothermal amplification techniques, CRISPR-Cas-based biosensors, nanotechnology-enhanced platforms like surface-enhanced Raman scattering (SERS) and nanoparticle-assisted hybridization, as well as microfluidic lab-on-a-chip systems, offer enhanced signal amplification, real-time detection, and high-throughput capabilities [13,14,15]. While some of these cutting-edge platforms provide superior performance, they are often less accessible and more expensive, limiting routine clinical use. Lateral flow assays and electrochemical biosensors offer a balance of cost-effectiveness, rapid detection, and minimal sample requirements, making them promising tools for point-of-care applications. Collectively, these emerging techniques are expanding the potential for accurate, multiplexed miRNA detection in the near future, supporting early diagnosis and personalized medicine.

## 2. Materials and Methods

This investigation was carried out following the rules of the Declaration of Helsinki. This study was approved by the Bioethics Committee at the Medical University of Warsaw (No KB/179/2020). All patients gave informed consent for participation in the study.

### 2.1. Tissue Samples

Tissue samples were prospectively collected from 28 patients with suspicious hypertrophic changes in the vocal fold mucosa, forming the study group. Inclusion criteria required a primary laryngeal lesion with no prior history of laryngeal surgery or radiotherapy in the neck region. Patients were excluded if they had advanced laryngeal tumors impairing vocal fold mobility (clinical stage T3 or higher), a history of other cancers, chemotherapy, or treatment with biologic drugs.

Additionally, tissue samples were collected from three patients with benign post-traumatic vocal fold lesions (polyps), who served as the control group. Exclusion criteria for the control group included a history of smoking or any oncological condition.

Informed consent was obtained from each participant prior to scheduled surgical resection, performed via transoral laryngeal microsurgery. A fragment of the pathological tissue (~3 × 3 mm) was excised, preserved in RNAlater™ Stabilization Solution (Invitrogen, Waltham, MA, USA), and stored at −80 °C until RNA isolation. The remaining tissue was marked at the biopsy site and submitted for histopathological analysis. Final diagnoses were categorized into four histopathological groups: no dysplasia (ND), low-grade dysplasia (LGD), high-grade dysplasia (HGD), and invasive cancer (IC). Tissue collection continued until seven patients were included in each diagnostic category. Once all samples were acquired, molecular analysis was conducted simultaneously.

### 2.2. Molecular Analysis

Molecular analyses were conducted at the Department of Applied Pharmacy. Total RNA, including the miRNA fraction, was extracted using the mirVana™ miRNA Isolation Kit (Ambion; Thermo Fisher Scientific, Waltham, MA, USA) in accordance with the manufacturer’s guidelines. The concentration and integrity of the isolated RNA were evaluated with a Q5000 spectrophotometer (Quawell Technology, San Jose, CA, USA).

For miRNA expression profiling, TaqMan™ Advanced miRNA Human A Cards (Applied Biosystems, Foster City, CA, USA) were employed, allowing simultaneous quantification of 381 miRNAs. Complementary DNA (cDNA) was generated using the TaqMan™ Advanced miRNA cDNA Synthesis Kit (Applied Biosystems) following the supplier’s protocol. Quantitative real-time PCR was subsequently carried out with TaqMan™ Fast Advanced Master Mix on a ViiA™ Real-Time PCR System (Applied Biosystems) under the recommended cycling conditions. Each sample was processed in a single technical run.

Data were processed using Expression Suite version 1.3 (Thermo Fisher Scientific), with a threshold set at 0.1 for all samples. Only miRNAs with Ct values < 32 were included in the analysis. miRNA expression levels were reported as ΔCt (dCt) values, normalized to the endogenous control hsa-miR-16-5p.

### 2.3. Statistical Analysis

Statistical analysis was conducted using Statistica v.13.3 (StatSoft Polska Sp. z o.o., Kraków, Poland). A *p*-value < 0.05 was considered statistically significant. Results are presented as means with 95% confidence intervals (CIs). Analysis of variance (ANOVA) was used to compare dCt values across groups. For group-wise comparisons, either the Student’s *t*-test (for normally distributed data) or the Wilcoxon signed-rank test (for non-normal distributions) was applied. Fold change was calculated using the formula: 2^−ΔΔCt^ = 2^−(dCt_sample − dCt_calibrator)^. Heat maps were generated based on 2^−dCt^ values for specific miRNA types across the analyzed groups.

## 3. Results

The quality of miRNA profiling was acceptable for all included tissue samples. The demographic and clinical characteristics of the study group—sex, age, and number of patients per final histopathological diagnosis—are summarized in Table 1.

Among the 381 miRNA types included in the TaqMan™ Advanced miRNA Human A Card, we detected 254 unique miRNAs in our cohort. Two miRNAs, hsa-miR-216a-5p-477976_mir and hsa-miR-488-3p-478129_mir, were selectively present only in the control group, with no expression observed in any hypertrophic laryngeal lesion. Conversely, hsa-miR-105-5p-477865_mir and hsa-miR-516a-5p-478978_mir were detected exclusively in high-grade dysplasia (HGD) and invasive cancer (IC) samples. These miRNAs were absent in the control group and in less advanced laryngeal lesions. In addition, hsa-miR-212-3p-478318_mir and hsa-miR-548d-5p-480870_mir were observed only in samples with IC, HGD, or low-grade dysplasia (LGD) and not in control or non-dysplastic (ND) samples. The miRNAs highlighted in Table 2 and discussed in the text were not consistently expressed across multiple groups and did not demonstrate statistically significant differences in the specific comparisons presented. Stage- or group-specific miRNAs are mentioned as potentially biologically important biomarkers.

Comparative dCt analysis between groups (control, ND, LGD, HGD, and IC) was performed using parametric or nonparametric tests depending on data distribution. Statistically significant differences in dCt values were observed for 121 miRNAs across these groups.

### 3.1. miRNA Expression: Control vs. ND, LGD, HGD, and IC

Given the clinical importance of distinguishing normal mucosa from lesions with dysplasia or cancer, we compared mean dCt values between the control group and each histopathological category. Statistically significant differences (*p* < 0.05) are shown in Table 3, and corresponding miRNA expression levels are visualized in Figure 1.

Most miRNAs showed significant differences only for one comparison with the control group, suggesting limited diagnostic value. However, hsa-miR-330-3p-478030_mir and hsa-miR-455-5p-478113_mir were significantly upregulated across three comparisons: control vs. LGD, HGD, and IC. These miRNAs may serve as potential markers for progression from normal mucosa to dysplasia or cancer.

Additionally, hsa-miR-100-5p-478224_mir, hsa-miR-125b-5p-477885_mir, hsa-miR-21-5p-477975_mir, hsa-miR-375-478074_mir, hsa-miR-629-5p-478183_mir, hsa-miR-708-5p-478197_mir, and hsa-miR-93-5p-478210_mir were significantly dysregulated in both HGD and IC compared to controls. Among them, hsa-miR-100-5p, hsa-miR-125b-5p, and hsa-miR-375 showed increased expression correlating with higher histological grade.

Fold changes in significantly dysregulated miRNAs are illustrated in Figure 2. Notably, the highest number of dysregulated miRNAs (n = 38) was observed when comparing the control group with IC, with 25 miRNAs specific to the IC stage.

### 3.2. miRNA Expression: ND vs. LGD, HGD, and IC

We next compared miRNA profiles between non-dysplastic lesions (ND) and those with LGD, HGD, or IC to identify markers indicative of dysplasia or malignancy. Hsa-miR-185-5p-477939_mir and hsa-miR-21-5p-477975_mir were significantly dysregulated with increasing histological severity and were the only two miRNAs that differed across all three comparisons (ND vs. LGD, ND vs. HGD, and ND vs. IC). Hsa-miR-21-5p expression increased from LGD (2.3) to HGD (2.99) but decreased in IC (1.3). In contrast, hsa-miR-185-5p showed a continuous rise from LGD (1.41) through HGD (1.85) to IC (2.04), indicating progressively higher expression with lesion severity. Expression trends are shown in Figure 3, and corresponding statistical results are shown in Table 4. Fold change data are presented in Figure 4.

### 3.3. miRNA Expression: LGD vs. HGD, LGD vs. IC, and HGD vs. IC

To explore miRNA-based differentiation between advanced lesion grades, we compared LGD vs. HGD, LGD vs. IC, and HGD vs. IC. The results are presented in Table 5.

A total of 25 miRNAs showed significantly different dCt values in both LGD vs. HGD and LGD vs. IC comparisons, suggesting potential as indicators of LGD stage. In contrast, hsa-miR-320a-478594_mir showed a significant difference only on comparison between HGD and IC, while 11 miRNAs were significantly altered both in this comparison and between LGD vs. IC and HGD vs. IC. Heatmaps of significant comparisons are shown in Figure 5 and Figure 6, while fold changes are illustrated in Figure 7A–D.

## 4. Discussion

MicroRNAs (miRNAs) have gained considerable attention as potential biomarkers for cancer diagnosis and treatment. Numerous investigations have demonstrated altered miRNA expression patterns in head and neck squamous cell carcinomas (HNSCCs), including laryngeal squamous cell carcinoma (LSCC), and assessed their utility in early detection as well as prognostic evaluation [9,10,11,12]. The development of LSCC is generally recognized as a multistep process, beginning with histologically normal epithelium and progressing through low-grade dysplasia (LGD) and high-grade dysplasia (HGD), ultimately culminating in invasive carcinoma (IC). This progression is accompanied by a gradual accumulation of genomic alterations, including copy number variations, somatic mutations, and gene fusions. In recent years, epigenetic mechanisms, such as DNA methylation and the regulatory actions of non-coding RNAs like miRNAs, have been recognized as critical contributors to this process. To date, most studies examining miRNA expression in LSCC have been limited to selected miRNAs. A comprehensive profiling of miRNA alterations across the full histopathological spectrum of laryngeal lesions has not yet been conducted. This study aimed to address that gap. Initial microarray-based profiling remains the gold standard for assessing miRNA expression patterns. In gastrointestinal malignancies such as colorectal, esophageal, and gastric adenocarcinomas, well-characterized miRNA profiles have demonstrated diagnostic utility for early cancer detection and potential as therapeutic targets [13,14]. A similar approach may be applicable to LSCC.

In our cohort, we found miR-375 to be the most significantly dysregulated molecule, with expression levels more than 30-fold higher in various stages of dysplasia and cancer compared to control tissues. This aligns partially with findings by Wu et al., who identified elevated levels of miR-148a and miR-375 across dysplastic stages of LSCC [10]. Notably, miR-375 overexpression has been reported in several cancers, including prostate, renal, and squamous cell carcinoma of the skin, where it has been associated with tumor progression and metastatic potential [15,16,17]. Robison et al. also linked miR-375 upregulation to lymphovascular invasion and aggressive histological features in SCC [18]. Overexpression of miR-375 has been implicated in cancer progression and metastasis through diverse mechanisms. It promotes aggressive behavior in NSCLC and prostate cancer via activation of ERK/MYC and STAT3 signaling [19,20], respectively, while in medullary thyroid carcinoma, it correlates with nodal spread through YAP1 suppression [21]. In Merkel cell carcinoma, exosomal miR-375 reprograms fibroblasts by targeting RBPJ/TP53, fostering a pro-metastatic niche [22], and in ER-positive breast cancer, it sustains ERα activity by downregulating RASD1 [23]. These findings suggest that miR-375 functions as an oncomiR capable of modulating both tumor-intrinsic pathways and the microenvironment, supporting its utility as a biomarker of aggressiveness and a potential therapeutic target. Our results further reinforce the potential role of miR-375 as a biomarker in early laryngeal carcinogenesis.

Similarly, our analysis supported the involvement of miR-21-5p, a well-established oncogenic miRNA, in laryngeal carcinogenesis. We observed significantly elevated expression in HGD and IC when compared to both the control and ND groups. These findings are consistent with previous studies, such as Wei et al., who reported elevated miR-21 levels in both premalignant and malignant laryngeal tissues [11]. While miR-21 is not specific to LSCC, its progressive increase with advancing pathology suggests a role in monitoring disease progression or recurrence. Overexpression of miR-21-5p has been widely implicated in the progression and aggressiveness of multiple cancers. It promotes tumor growth and survival primarily through suppression of PTEN, resulting in PI3K/AKT pathway activation [24], and by targeting PDCD4, which enhances AP-1-mediated transcription of genes driving proliferation and invasion [25]. Additionally, miR-21-5p facilitates epithelial-to-mesenchymal transition via SMAD7 inhibition and promotes extracellular matrix remodeling through downregulation of RECK, leading to increased MMP2/9 activity [26]. By also modulating apoptosis- and cell cycle-related proteins such as BCL2, TPM1, and SPRY2, miR-21-5p contributes to enhanced metastatic potential and chemoresistance [26].

In contrast to previous studies, such as Tuncturk et al., we found only miR-155-5p to be significantly altered (control vs. LGD), while miR-183-5p and miR-106b-3p showed no significant expression differences in our dataset [12]. The discrepancy may be due to differences in profiling panels or sample sizes.

Among markers that distinguished early from advanced lesions, miR-143-3p showed the most remarkable change, with expression levels increasing over 25-fold from LGD to HGD and IC. This contrasts with previous reports identifying miR-143 as a tumor suppressor in advanced LSCC and warrants further investigation [27,28]. A similar pattern was seen with miR-26a-5p, another tumor suppressor typically downregulated in other malignancies [29,30,31]. In our data, miR-26a-5p expression was strongly upregulated in HGD and IC compared to LGD. Its known role in regulating apoptosis and autophagy, and in suppressing tumor proliferation and metastasis, suggests that context-dependent factors may influence its function in early laryngeal carcinogenesis stages.

Another miRNA of interest is miR-145-5p, which was upregulated nearly 4.4-fold in IC compared to LGD. Given its involvement in regulating genes linked to proliferation, differentiation, and apoptosis, further research into its role and regulatory targets in laryngeal cancer is justified [32,33].

On the other hand, miR-126-3p was significantly downregulated with increasing dysplasia severity and invasive disease. This trend is consistent with prior studies associating low miR-126-3p levels with advanced LSCC, poor differentiation, and metastasis [7,34,35]. This miRNA likely acts as a tumor suppressor, and its restoration may offer therapeutic potential.

Strikingly, miR-216a-5p and miR-488-3p were exclusively detected in control tissues and absent in all pathological samples, suggesting their potential role as tumor suppressors lost during early carcinogenesis. This is supported by literature indicating that both miRNAs inhibit tumor-related pathways and regulate apoptosis, migration, and cell cycle progression [36,37].

Conversely, miR-105-5p and miR-516a-5p were expressed only in HGD and IC, but not in less advanced lesions or normal tissues. MiR-105-5p, in particular, has been implicated in EMT activation, metastasis, and chemoresistance in multiple cancers [38,39]. Their selective expression in advanced laryngeal lesions supports a role in malignant transformation and progression.

Interestingly, hsa-miR-503-5p showed an upward trend in expression with increasing histological severity of dysplasia and progression to laryngeal carcinoma. TargetScan predictions and literature curation identify several high-confidence targets of hsa-miR-503-5p, including the tumor suppressor *PDCD4* and key regulators of the cell cycle and apoptosis (*CCND1/CCNE1/CDC25A*, *PUMA*, *SMAD7*, and *TIMP2*) [40,41]. In our cohort, elevated miR-503-5p expression would be expected to suppress these critical brakes, thereby promoting proliferative expansion, extracellular matrix remodeling and invasion, as well as evasion of apoptosis—hallmarks that parallel the progression from dysplasia to carcinoma. Taken together, these findings support a model in which miR-503-5p upregulation drives disease progression primarily through *PDCD4* loss, further reinforced by dysregulation of cell-cycle and survival pathways.

Given the limited available data on miRNA profiles across LSCC stages, our study employed the TaqMan™ Advanced miRNA Human A Card, a broad-spectrum profiling tool, to enable comprehensive analysis. We believe that using such a wide panel is essential for reliable marker discovery.

In future studies, validation of the identified miRNAs in larger patient cohorts and their assessment in minimally invasive samples, such as saliva or serum, will be critical for clinical application. Moreover, elucidating the functional roles of these miRNAs and their downstream targets may enhance our understanding of laryngeal tumorigenesis and open new opportunities for therapeutic intervention.

We are committed to continuing this line of research and hope that sharing our findings will stimulate further studies. The identification of reliable miRNA markers has the potential to significantly improve early detection, risk stratification, and individualized treatment planning for patients with laryngeal lesions.

## 5. Conclusions

This study provides a comprehensive overview of miRNA expression profiles across various stages of laryngeal squamous cell carcinoma progression, from non-dysplastic tissue to invasive carcinoma. Using a broad-spectrum microarray platform, we identified several miRNAs with stage-specific expression patterns that may serve as potential biomarkers for early detection and progression monitoring in LSCC.

Key findings include the significant upregulation of miR-375, miR-143-3p, miR-145-5p, and miR-26a-5p in more advanced dysplastic lesions and the progressive downregulation of miR-126-3p. Additionally, miR-216a-5p and miR-488-3p were exclusively present in normal tissues, suggesting their possible role as tumor suppressors lost during early tumorigenesis, whereas miR-105-5p and miR-516a-5p were selectively expressed in high-grade and invasive lesions.

These findings support the hypothesis that miRNAs play a pivotal role in the multistep transformation of laryngeal epithelium and may serve as valuable biomarkers for diagnosis, prognosis, and potential therapeutic targeting.

Further validation in larger patient cohorts and exploration of circulating miRNA expression in saliva or blood samples are needed to confirm their clinical utility and applicability in non-invasive screening strategies.

## Figures and Tables

**Figure 1 cancers-17-02883-f001:**
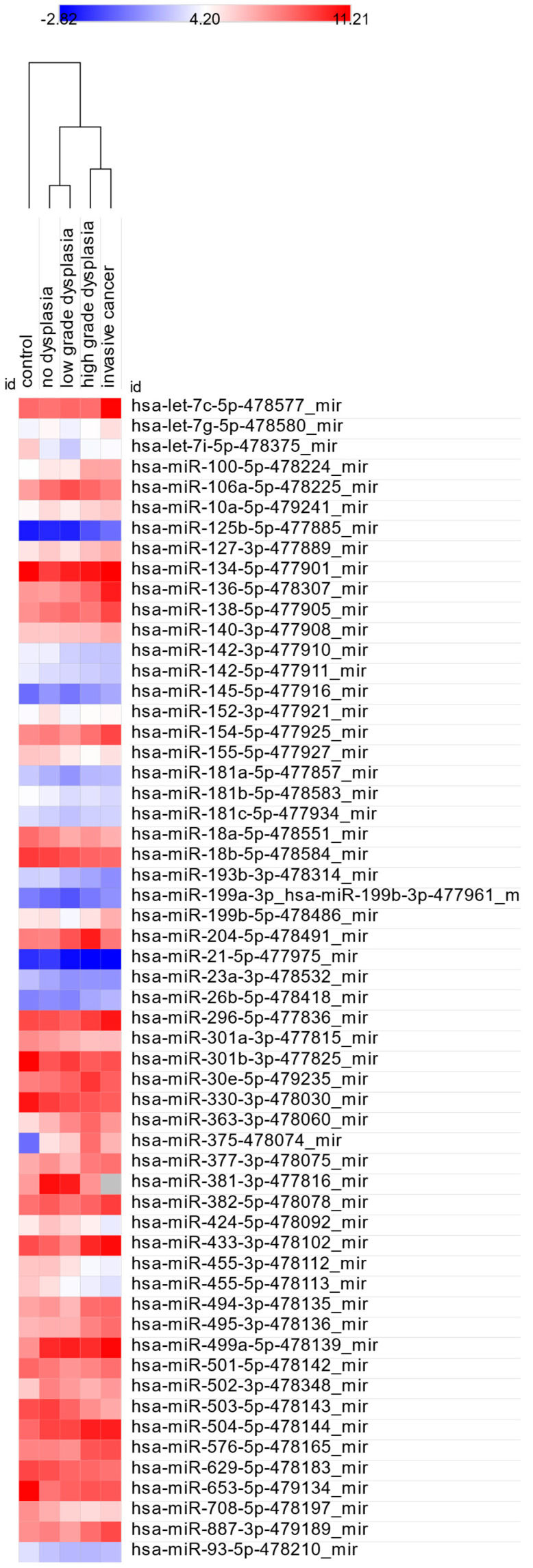
Heatmap of significantly dysregulated miRNAs across control, ND, LGD, HGD, and IC samples.

**Figure 2 cancers-17-02883-f002:**
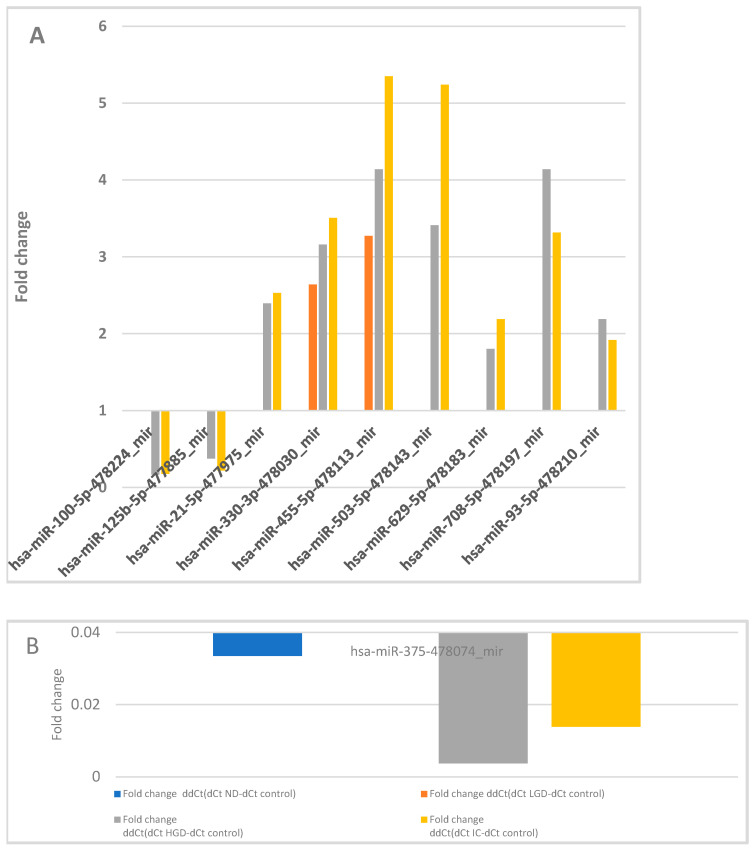
Fold changes in significantly dysregulated miRNAs between the control group and lesion categories ((**A**,**B**) graphical presentation of selected significantly dysregulated miRNA types; (**C**) numerical values for the analyzed types). ns—mean dCt not significant on comparison; nr—not relevant.

**Figure 3 cancers-17-02883-f003:**
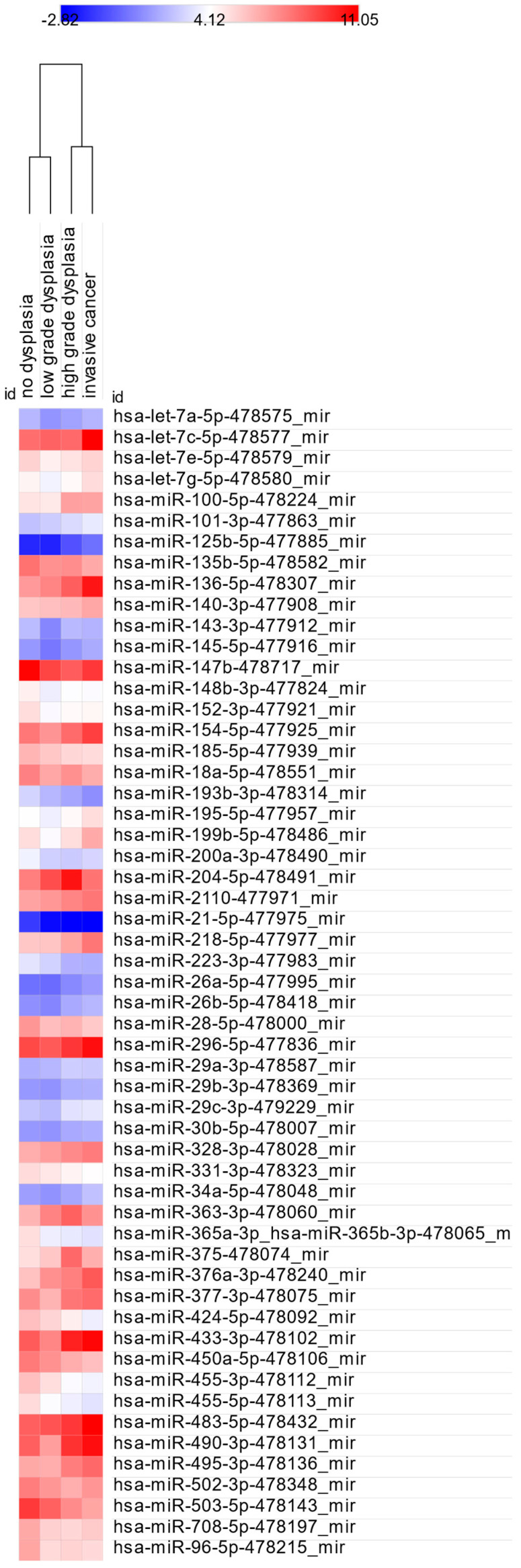
Heatmap of miRNAs significantly differing between ND and dysplastic/cancerous lesions.

**Figure 4 cancers-17-02883-f004:**
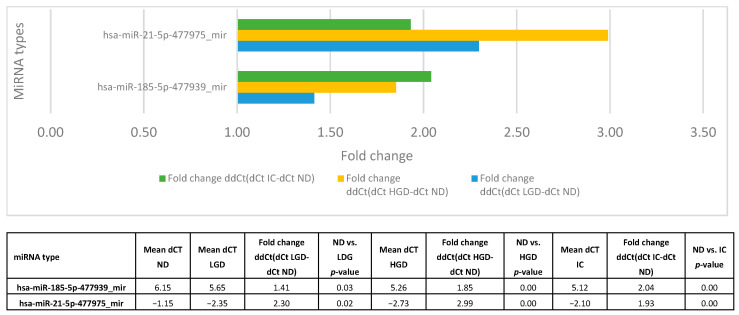
Fold changes for hsa-miR-185-5p and hsa-miR-21-5p across lesion categories.

**Figure 5 cancers-17-02883-f005:**
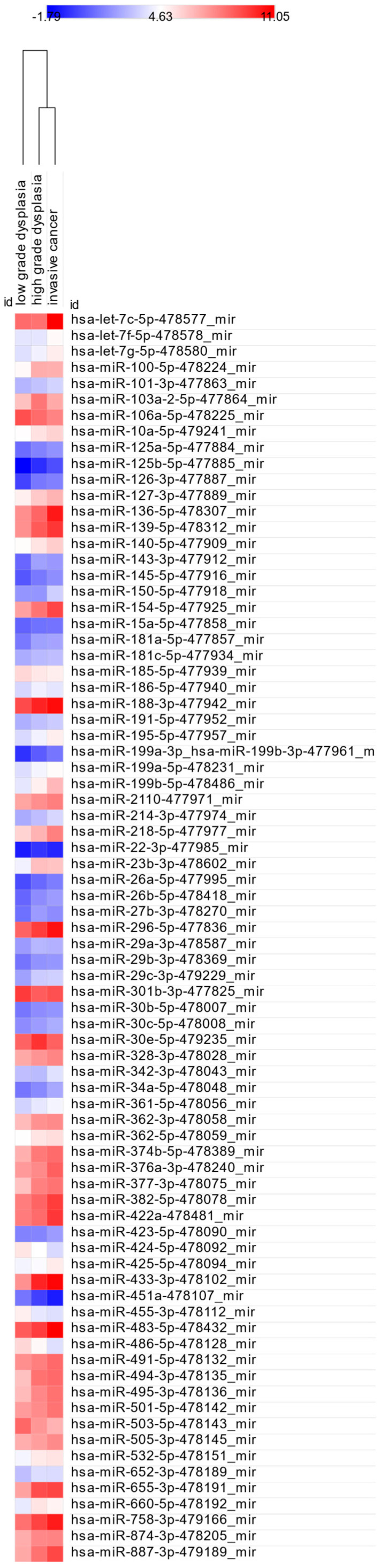
Heatmap of miRNA expression differences between LGD vs. HGD and LGD vs. IC.

**Figure 6 cancers-17-02883-f006:**
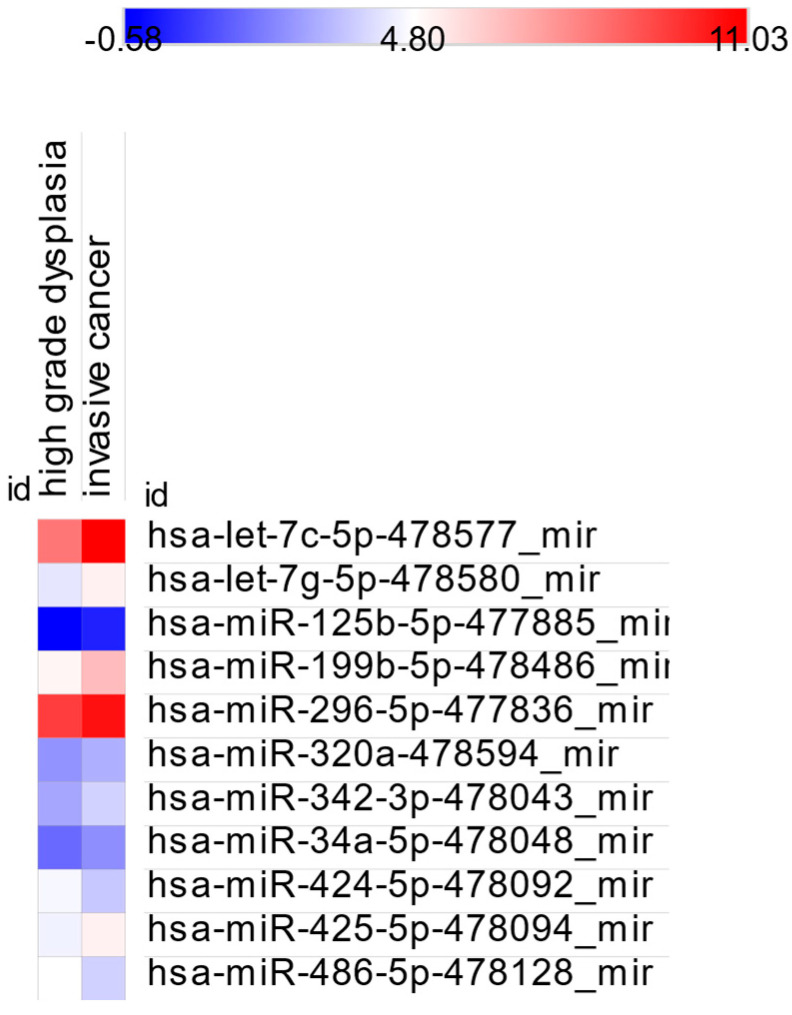
Heatmap of miRNA expression differences between HGD and IC.

**Figure 7 cancers-17-02883-f007:**
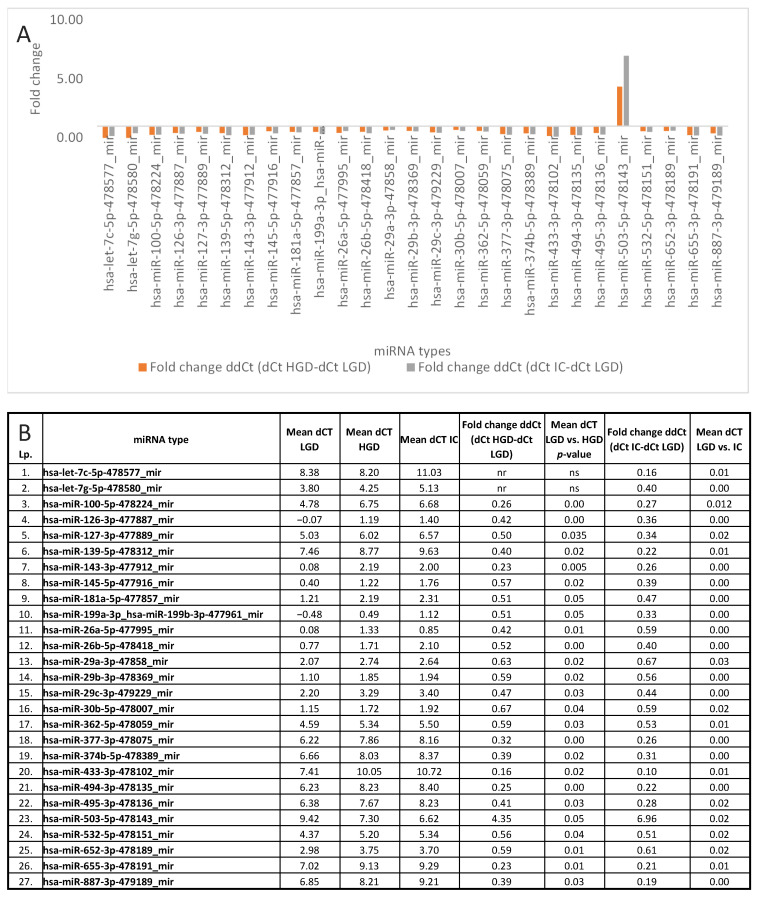
Fold change in miRNA expression levels across lesion progression from low-grade dysplasia (LGD) to invasive cancer (IC): (**A**) graphical presentation of fold changes for significantly different miRNA types on comparison between LGD vs. HGD and LGD vs. IC; (**B**) numerical values for the analyzed miRNA types; (**C**) graphical presentation of fold changes for significantly different miRNA types on comparison between HGD vs. IC; (**D**) numerical values for the analyzed miRNA types.

**Table 1 cancers-17-02883-t001:** Characteristics of the study group.

	No	Mean Age (Years) ±SD	Sex (M/F)
All patients	31	64.16 ± 10.43	20/11
Control group	3	48.67 ± 6.55	2/1
No dysplasia (ND)	7	70.43 ± 6.3	1/6
Low-grade dysplasia (LGD)	7	59.14 ± 8.48	5/2
High-grade dysplasia (HGD)	7	66.57 ± 10.2	6/1
Invasive cancer (IC)	7	67.14 ± 8.13	6/1
T1	5	66.8 ± 5.77	4/1
T2	2	70.5 ± 11.5	2/0

**Table 2 cancers-17-02883-t002:** Potentially predictive miRNAs identified across specific histopathological categories.

**miRNA Identified only in control group**	**No of samples with detected miRNA Type**	**Mean dCt**
hsa-miR-208b-3p-477806_mir	1	5.82
hsa-miR-216a-5p-477976_mir	2	10.83
hsa-miR-133b-480871_mir	1	4.98
hsa-miR-499a-3p-478948_mir	1	9.26
hsa-miR-876-5p-479187_mir	1	9.70
hsa-miR-208a-3p-477819_mir	1	8.48
hsa-miR-488-3p-478129_mir	2	10.11
**miRNA identified only in IC and HGD patients**	**No of samples with detected miRNA type**	**Mean dCt**
hsa-miR-105-5p-477865_mir	6	9.37
hsa-miR-516a-5p-478978_mir	6	10.11
**miRNA identified only in IC, HGD, and LGD patients**	**No of samples with detected miRNA type**	**Mean dCt**
hsa-miR-212-3p-478318_mir	10	10.13
hsa-miR-548d-5p-480870_mir	11	10.63

**Table 3 cancers-17-02883-t003:** Significant (*p* < 0.05) differences in dCt values between the control group and laryngeal lesion categories.

miRNA Type	Control vs. ND *p*-Value	Control vs. LGD *p*-Value	Control vs. HGD *p*-Value	Control vs. IC *p*-Value
hsa-let-7c-5p-478577_mir				0.029
hsa-let-7g-5p-478580_mir				0.007
hsa-let-7i-5p-478375_mir		0.04		
**hsa-miR-100-5p-478224_mir**			0.002	0.012
hsa-miR-106a-5p-478225_mir		0.001		
hsa-miR-10a-5p-479241_mir				0.027
**hsa-miR-125b-5p-477885_mir**			0.015	0.003
hsa-miR-127-3p-477889_mir				0.024
hsa-miR-134-5p-477901_mir		0.044		
hsa-miR-136-5p-478307_mir				0.014
hsa-miR-138-5p-477905_mir				0.03
hsa-miR-140-3p-477908_mir				0.029
hsa-miR-142-3p-477910_mir				0.023 ^#^
hsa-miR-142-5p-477911_mir				0.011
hsa-miR-145-5p-477916_mir				0.046
hsa-miR-152-3p-477921_mir	0.036 ^#^			
hsa-miR-154-5p-477925_mir				0.016
hsa-miR-155-5p-477927_mir		0.031		
hsa-miR-181a-5p-477857_mir		0.028		
hsa-miR-181b-5p-478583_mir		0.028		0.035
hsa-miR-181c-5p-477934_mir		0.028		
hsa-miR-18a-5p-478551_mir		0.03		0.027
hsa-miR-18b-5p-478584_mir				0.032
hsa-miR-193b-3p-478314_mir				0.036 ^#^
hsa-miR-199a-3p_hsa-miR-199b-3p-477961_mir		0.045		
hsa-miR-199b-5p-478486_mir				0.02
hsa-miR-204-5p-478491_mir			0.048	
**hsa-miR-21-5p-477975_mir**			0.037	0.034
hsa-miR-23a-3p-478532_mir			0.014	
hsa-miR-26b-5p-478418_mir				0.021
hsa-miR-296-5p-477836_mir				0.048
hsa-miR-301a-3p-477815_mir				0.046
hsa-miR-301b-3p-477825_mir			0.002	
hsa-miR-30e-5p-479235_mir		0.038	0.039	
**hsa-miR-330-3p-478030_mir**		0.033	0.016	0.019
hsa-miR-363-3p-478060_mir		0.013	0.004	
**hsa-miR-375-478074_mir**	0.018		0.002	0.011
hsa-miR-377-3p-478075_mir				0.042
hsa-miR-381-3p-477816_mir		0.048		
hsa-miR-382-5p-478078_mir				0.038
hsa-miR-424-5p-478092_mir	0.013			0.038
hsa-miR-433-3p-478102_mir				0.007
hsa-miR-455-3p-478112_mir				0.040
**hsa-miR-455-5p-478113_mir**		0.032	0.047	0.047
hsa-miR-494-3p-478135_mir			0.031	
hsa-miR-495-3p-478136_mir			0.04 ^#^	
has-miR-499a-5p-478139_mir				0.041
hsa-miR-501-5p-478142_mir		0.028		
hsa-miR-502-3p-478348_mir	0.023 ^#^			
**hsa-miR-503-5p-478143_mir**			0.002	0.012
hsa-miR-504-5p-478144_mir		0.019		
hsa-miR-576-5p-478165_mir				0.029
**hsa-miR-629-5p-478183_mir**			0.018	0.007
hsa-miR-653-5p-479134_mir				0.048
**hsa-miR-708-5p-478197_mir**			0.030	0.042
hsa-miR-887-3p-479189_mir				0.026
**hsa-miR-93-5p-478210_mir**			0.016	0.040 ^#^
Total number of identified significantly different miRNAs	**4**	**16**	**17**	**38**

^#^ Wilcoxon test, other with *t*-test. Those **bolded** are miRNA types with at least two significantly different levels of dCt on comparison.

**Table 4 cancers-17-02883-t004:** Significant differences in dCt values between ND and LGD, HGD, and IC.

miRNA Type	ND vs. LGD	ND vs. HGD	ND vs. IC
hsa-let-7a-5p-478575_mir	0.045		
hsa-let-7c-5p-478577_mir			0.006
hsa-let-7e-5p-478579_mir	0.03 ^#^		
hsa-let-7g-5p-478580_mir	0.029		0.016
hsa-miR-100-5p-478224_mir		0.003	0.012
hsa-miR-101-3p-477863_mir			0.013
hsa-miR-125b-5p-477885_mir		0.021 ^#^	0.005 ^#^
hsa-miR-135b-5p-478582_mir			0.017
hsa-miR-136-5p-478307_mir			0.007
hsa-miR-140-3p-477908_mir			0.018
hsa-miR-143-3p-477912_mir	0.002		
hsa-miR-145-5p-477916_mir	0.045		
hsa-miR-147b-478717_mir		0.021	
hsa-miR-148b-3p-477824_mir	0.018		
hsa-miR-152-3p-477921_mir	0.032		
hsa-miR-154-5p-477925_mir			0.014
**hsa-miR-185-5p-477939_mir**	0.026	0.001	0.002 ^#^
hsa-miR-18a-5p-478551_mir	0.038		0.026
hsa-miR-193b-3p-478314_mir			0.045
hsa-miR-195-5p-477957_mir			0.042
hsa-miR-199b-5p-478486_mir			0.003
hsa-miR-200a-3p-478490_mir		0.035	
hsa-miR-204-5p-478491_mir		0.026	
**hsa-miR-21-5p-477975_mir**	0.021	0.002	0.001
hsa-miR-2110-477971_mir			0.021
hsa-miR-218-5p-477977_mir			0.01
hsa-miR-223-3p-477983_mir		0.049	0.041
hsa-miR-26a-5p-477995_mir		0.017	0.001
hsa-miR-26b-5p-478418_mir		0.023	0.001
hsa-miR-28-5p-478000_mir	0.01		0.004
hsa-miR-296-5p-477836_mir			0.01
hsa-miR-29a-3p498587_mir		0.021	0.026
hsa-miR-29b-3p-478369_mir			0.017
hsa-miR-29c-3p-479229_mir			0.018
hsa-miR-30b-5p-478007_mir			0.044
hsa-miR-328-3p-478028_mir			0.025
hsa-miR-331-3p-478323_mir			0.037
hsa-miR-34a-5p-478048_mir			0.03
hsa-miR-363-3p-478060_mir	0.036	0.003	
hsa-miR-365a-3p_hsa-miR-365b-3p-478065_mir	0.15 ^#^		0.041 ^#^
hsa-miR-375-478074_mir		0.021	
has-miR-376a-3p-478240_mir			0.007
hsa-miR-377-3p-478075_mir	0.041 ^#^		
hsa-miR-424-5p-478092_mir		0.007	0.000.
hsa-miR-433-3p-478102_mir			0.020
hsa-miR-450a-5p-478106_mir			0.020
hsa-miR-455-3p-478112_mir		0.011	0.006
hsa-miR-455-5p-478113_mir		0.049	0.031
hsa-miR-483-5p-478432_mir			0.010
hsa-miR-490-3p-478131_mir			0.037
hsa-miR-495-3p-478136_mir			0.023 ^#^
hsa-miR-502-3p-478348_mir		0.049	
hsa-miR-503-5p-478143_mir		0.018	0.003
hsa-miR-708-5p-478197_mir		0.027	
hsa-miR-96-5p-478215_mir	0.015 ^#^		0.03 ^#^
Total number of identified significantly different miRNAs	**16**	**18**	**40**

^#^ Wilcoxon test, other with *t*-test. Those **bolded** are miRNA types with three significantly different levels of dCt on comparison.

**Table 5 cancers-17-02883-t005:** Significantly different miRNAs in comparisons: LGD vs. HGD, LGD vs. IC, and HGD vs. IC.

miRNA Type	LGD vs. HGD	LGD vs. IC	HGD vs. IC
hsa-let-7c-5p-478577_mir		0.009	0.032
hsa-let-7f-5p-478578_mir		0.04	
hsa-let-7g-5p-478580_mir		0.001	0.002 ^#^
hsa-miR-100-5p-478224_mir	0.004	0.012	
hsa-miR-101-3p-477863_mir		0.011	
hsa-miR-103a-2-5p-477864_mir	0.026		
has-miR-106a-5p-478225_mir		0.021	
hsa-miR-10a-5p-479241_mir		0.003	
hsa-miR-125a-5p-477884_mir		0.002	
hsa-miR-125b-5p-477885_mir			0.047
hsa-miR-126-3p-477887_mir	0.002	0.002	
hsa-miR-127-3p-477889_mir	0.035	0.02	
hsa-miR-136-5p-478307_mir		0.007	
hsa-miR-139-5p-478312_mir	0.015	0.01	
hsa-miR-140-5p-477909_mir		0.041 ^#^	
hsa-miR-143-3p-477912_mir	0.005	0.001	
hsa-miR-145-5p-477916_mir	0.017	0.003	
hsa-miR-150-5p-477918_mir		0.037	
hsa-miR-154-5p-477925_mir		0.001	
hsa-miR-15a-5p-477858_mir		0.049	
hsa-miR-181a-5p-477857_mir	0.046	0.010	
hsa-miR-181c-5p-477934_mir		0.049	
hsa-miR-185-5p-477939_mir		0.021 ^#^	
hsa-miR-186-5p-477940_mir	0.028		
hsa-miR-188-3p-477942_mir		0.009	
hsa-miR-191-5p-477952_mir		0.007	
hsa-miR-195-5p-477957_mir		0.04	
hsa-miR-199a-5p-478231_mir		0.019	
hsa-miR-199a-3p_hsa-miR-199b-3p-477961_mir	0.045	0.004	
hsa-miR-199b-5p-478486_mir			0.026
hsa-miR-214-3p-477974_mir		0.027	
hsa-miR-2110-477971_mir		0.002	
hsa-miR-218-5p-477977_mir		0.012	
hsa-miR-22-3p-477985_mir	0.046		
hsa-miR-23b-3p-478602_mir	0.036		
hsa-miR-26a-5p-477995_mir	0.009	0.001	
hsa-miR-26b-5p-478418_mir	0.004	0.001	
hsa-miR-27b-3p-478270_mir	0.006		
hsa-miR-296-5p-477836_mir			0.04
hsa-miR-29a-3p-47858_mir	0.022	0.025	
hsa-miR-29b-3p-478369_mir	0.023	0.002	
hsa-miR-29c-3p-479229_mir	0.027	0.004	
hsa-miR-301b-3p-477825_mir	0.020 ^#^		
hsa-miR-30b-5p-478007_mir	0.036	0.015	
hsa-miR-30c-5p-478008_mir		0.012	
hsa-miR-30e-5p-479235_mir	0.032		
hsa-miR-320a-478594_mir			0.048
hsa-miR-328-3p-478028_mir		0.037	
hsa-miR-342-3p-478043_mir		0.02	0.032
hsa-miR-376a-3p-478240_mir		0.041	
hsa-miR-361-5p-478056_mir		0.033	
hsa-miR-362-3p-478058_mir		0.038	
hsa-miR-362-5p-478059_mir	0.032	0.014	
hsa-miR-374b-5p-478389_mir	0.015 ^#^	0.002 ^#^	
hsa-miR-377-3p-478075_mir	0.004	0.001	
hsa-miR-382-5p-478078_mir		0.018	
hsa-miR-433-3p-478102_mir	0.02	0.005	
hsa-miR-422a-478481_mir		0.042	
hsa-miR-34a-5p-478048_mir			0.04 ^#^
hsa-miR-423-5p-478090_mir		0.044	
hsa-miR-424-5p-478092_mir			0.043
hsa-miR-425-5p-478094_mir		0.037	0.038
hsa-miR-455-3p-478112_mir		0.044	
hsa-miR-451a-478107_mir		0.042	
hsa-miR-483-5p-478432_mir		0.022	
hsa-miR-486-5p-478128_mir		0.024	0.046
hsa-miR-491-5p-478132_mir		0.035	
hsa-miR-494-3p-478135_mir	0.004	0.001	
hsa-miR-495-3p-478136_mir	0.026	0.015 ^#^	
hsa-miR-501-5p-478132_mir		0.023	
hsa-miR-503-5p-478143_mir	0.047	0.015	
hsa-miR-505-3p-478145_mir		0.011	
hsa-miR-532-5p-478151_mir	0.040	0.016	
hsa-miR-652-3p-478189_mir	0.005	0.020	
hsa-miR-655-3p-478191_mir	0.008	0.005	
hsa-miR-660-5p-478192_mir	0.041		
hsa-miR-758-3p-479166_mir		0.033	
hsa-miR-874-3p-478205_mir		0.034	
hsa-miR-887-3p-479189_mir	0.026	0.001	
Total number of identified significantly different miRNAs	33	65	11

^#^ Wilcoxon test, other with *t*-test.

## Data Availability

The data presented in this study are available on request from the corresponding authors.

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
