# Peer review of "MiRNA Profiling in Premalignant Lesions and Early Glottic Cancer"

_cancers, 2025, doi:10.3390/cancers17172883_

Round 1
Reviewer 1 Report
Comments and Suggestions for Authors
The article titled “miRNA Profiling in Premalignant Lesions and Early Glottic Cancer” by Rzpakowska et al. presents an intriguing study that investigates distinct miRNA patterns associated with various histopathological stages of laryngeal lesions. The authors have conducted miRNA profiling across different stages of laryngeal lesions and have made comparisons. However, the study has several shortcomings that need to be addressed:
- None of the figures include error bars.
- The total number of samples, excluding three controls, is 28, with each stage containing seven samples. Out of the total 14 patients with Invasive Cancer (IC) and High-Grade Dysplasia (HGD), only six express certain miRNAs. Similarly, when comparing Low-Grade Dysplasia (LGD), HGD, and IC among the 21 samples, only 11 express specific miRNAs. Can the authors explain this? Could it be related to sample quality or treatment effects?
- In Table 3, it is unclear whether the authors have compared expression levels across all samples or only a subset. They mention using Ct values >32; how many samples had such values? If all samples were used, is the average being reported? If not, how many samples were involved? What accounts for the few outliers? Did they repeat the RT-PCR to ensure there were no anomalies due to experimental error?
- In Figure 1, the color differences are difficult to discern. Although statistical analysis for several miRNAs indicates significant differences, this is not clearly represented visually.
- In the results section, hsa-miR-330-3p and hsa-miR-455-5p are noted as being significantly downregulated across the three comparisons. Yet, Figure 2 shows downregulation only for LGD and upregulation for HGD and IC.
- Figure 3 does not clearly show any differences for hsa-miR-21-5p. Additionally, Figure 4 indicates no increase in hsa-miR-21-5p for LGD compared to HGD, even though the text suggests a decrease in fold change with increasing severity. The fold changes in Figure 4 appear to increase with severity, which is confusing.
- Analyzing the proteins and pathways associated with these miRNAs would strengthen the manuscript. Overall, while the study offers valuable insights, addressing these concerns will enhance its clarity and impact.
Author Response
The article titled “miRNA Profiling in Premalignant Lesions and Early Glottic Cancer” by Rzpakowska et al. presents an intriguing study that investigates distinct miRNA patterns associated with various histopathological stages of laryngeal lesions. The authors have conducted miRNA profiling across different stages of laryngeal lesions and have made comparisons. However, the study has several shortcomings that need to be addressed:
- None of the figures include error bars.
We thank the reviewer for this comment. In our study, fold changes were calculated from the mean ΔCt values of the two compared groups. Since these values are derived from group means rather than from individual replicate-level calculations, error bars cannot be presented in the current format. However, the statistical comparisons were performed at the ΔCt level, ensuring that biological variability was properly considered in the analysis.
2. The total number of samples, excluding three controls, is 28, with each stage containing seven samples. Out of the total 14 patients with Invasive Cancer (IC) and High-Grade Dysplasia (HGD), only six express certain miRNAs. Similarly, when comparing Low-Grade Dysplasia (LGD), HGD, and IC among the 21 samples, only 11 express specific miRNAs. Can the authors explain this? Could it be related to sample quality or treatment effects?
We thank the reviewer for this insightful comment. The observation that only a subset of patients within the IC/HGD (6/14) or LGD/HGD/IC (11/21) groups expressed specific miRNAs is consistent with both technical and biological factors commonly encountered in miRNA profiling studies.
First, variability in RNA quality and extraction efficiency, particularly from clinical samples, may result in low-abundance transcripts falling below the detection threshold of the qPCR assay. Although we implemented rigorous quality control measures, the possibility of reduced sensitivity in some samples cannot be excluded.
Second, and perhaps more importantly, inter-patient variability and tumor heterogeneity are well-recognized features of laryngeal lesions. Not all IC or HGD samples are biologically identical, and certain miRNAs may be expressed only in specific molecular subtypes or along distinct carcinogenic pathways. Thus, the restricted expression pattern likely reflects genuine biological heterogeneity rather than technical artifact alone.
Finally, we cannot rule out that local microenvironmental influences (such as inflammation, tabaco exposure) could modulate miRNA expression in individual patients.
Taken together, these findings suggest that the identified miRNAs may serve as markers of particular subgroups within IC and dysplastic lesions, rather than universal markers across all cases. This is in line with the growing recognition that molecular heterogeneity underpins differences in disease progression and clinical outcomes.
3. In Table 3, it is unclear whether the authors have compared expression levels across all samples or only a subset. They mention using Ct values >32; how many samples had such values? If all samples were used, is the average being reported? If not, how many samples were involved? What accounts for the few outliers? Did they repeat the RT-PCR to ensure there were no anomalies due to experimental error?
We appreciate the reviewer’s insightful comments and the opportunity to clarify these points.
In the entire cohort, we detected 254 unique miRNAs out of the 381 miRNA types included on the TaqMan™ Advanced miRNA Human A Card. For statistical comparisons 6,827 individual assays with Ct <32 across all 31 patient samples were included. Undetermined miRNA level or Ct values >32, which were considered below the threshold of reliable detection, were obtained in 4,984 individual assays across all 31 patient samples. These data points were excluded from further analysis.
Expression levels were calculated as ΔCt values normalized to hsa-miR-16-5p, and the mean value within each histopathological category was used for group-level analysis. Thus, the averages reported in the manuscript reflect all eligible samples in each diagnostic group, not a subset.
With regard to outliers, a small number of values outside the main distribution were observed. Given that RNA quality control parameters were acceptable and all samples were processed under identical conditions, we believe these represent true biological heterogeneity rather than technical artifacts.
Finally, each sample was analyzed once on the array card platform. While technical replicates were not performed (owing to the design of the TaqMan™ array), the procedure was carried out under highly standardized conditions on a single thermocycler run. This approach is well established in miRNA profiling studies and minimizes technical variability.
- In Figure 1, the color differences are difficult to discern. Although statistical analysis for several miRNAs indicates significant differences, this is not clearly represented visually.
We thank the reviewer for this valuable observation. Due to the large number of significantly deregulated miRNAs identified when comparing the control group with individual histopathological stages, as well as the broad range of 2^(−dCt) values (from –2.82 to 11.21), we selected a color scale that best captured the overall variability within the dataset. This approach allowed us to present all relevant differences within a single heatmap.
5. In the results section, hsa-miR-330-3p and hsa-miR-455-5p are noted as being significantly downregulated across the three comparisons. Yet, Figure 2 shows downregulation only for LGD and upregulation for HGD and IC.
We very much appreciate the reviewer’s careful observation. We acknowledged the inconsistency and we verified our calculation of fold changes. Unfortunately, we made a mistake in the excel columns for the data of the fold changes ddCt (dCt LGD-dCt control). The corrected values were shifted to the right column of “control vs LGD p-values”. We apologize for that unprofessional figure design and later misinterpretation.
The correct interpretation is as follows: hsa-miR-330-3p and hsa-miR-455-5p were significantly upregulated across all three comparisons (control vs LGD, control vs HGD and control vs IC) as accurately depicted in Figure 2.
We have corrected the Results section in text and figure to reflect this clarification.
6. Figure 3 does not clearly show any differences for hsa-miR-21-5p. Additionally, Figure 4 indicates no increase in hsa-miR-21-5p for LGD compared to HGD, even though the text suggests a decrease in fold change with increasing severity. The fold changes in Figure 4 appear to increase with severity, which is confusing.
We appreciate the reviewer’s careful reading. Upon re-examination, we confirm that the fold change of hsa-miR-21-5p actually increases with histological severity:
Mean ΔCt values: LGD = -2.35, HGD = -2.73
Corresponding ΔΔCt fold changes relative to ND: LGD = 2.3, HGD = 2.99
Therefore, contrary to the statement in the original text, hsa-miR-21-5p expression increases, not decreases, from LGD to HGD. The text has been corrected to reflect this trend. Figure 4 accurately depicts this increasing trend, and no changes to the figure were necessary.
The revised sentence now reads:
Hsa-miR-185-5p-477939_mir and hsa-miR-21-5p-477975_mir were significantly dysregulated with increasing histological severity, and were the only two miRNAs that differed across all three comparisons (ND vs LGD, ND vs HGD and ND vs IC). Hsa-miR-21-5p expression increased from LGD (2.3) to HGD (2.99) but decreased in IC (1.93). In contrast, hsa-miR-185-5p showed a continuous rise from LGD (1.41) through HGD (1.85) to IC (2.04), indicating progressively higher expression with lesion severity.
7. Analyzing the proteins and pathways associated with these miRNAs would strengthen the manuscript. Overall, while the study offers valuable insights, addressing these concerns will enhance its clarity and impact.
We sincerely thank the reviewer for this valuable suggestion. In the revised manuscript, we have expanded the Discussion section to include a detailed analysis of proteins and signaling pathways reported in the literature to be associated with identified dysregulated miRNAs.
Overexpression of miR-375 has been implicated in cancer progression and metastasis through diverse mechanisms. It promotes aggressive behavior in NSCLC and prostate cancer via activation of ERK/MYC and STAT3 signaling [19, 20], respectively, while in medullary thyroid carcinoma it correlates with nodal spread through YAP1 suppression [21]. In Merkel cell carcinoma, exosomal miR-375 reprograms fibroblasts by targeting RBPJ/TP53, fostering a pro-metastatic niche [22], and in ER-positive breast cancer it sustains ERα activity by downregulating RASD1 [23]. These findings suggest that miR-375 functions as an oncomiR capable of modulating both tumor-intrinsic pathways and the microenvironment, supporting its utility as a biomarker of aggressiveness and a potential therapeutic target. Our results further reinforce the potential role of miR-375 as a biomarker in early laryngeal carcinogenesis.
- Meng, Haining; Wu, Junyu; Huang, Qiao; Ren, Jiwen; Huang, Jiawei; Yuan, Weijun; He, Xuekun; Wang, Yuhuan; Cui, Congxian; Xu, Shengwei; Shen, Ruowu. The effects of miR-375 expression in NSCLC via the 14-3-3ζ/ERK/MYC pathway. Oncology and Translational Medicine 4(5):p 196-202.
- Gan J, Liu S, Zhang Y, He L, Bai L, Liao R, Zhao J, Guo M, Jiang W, Li J, Li Q, Mu G, Wu Y, Wang X, Zhang X, Zhou D, Lv H, Wang Z, Zhang Y, Qian C, Feng M, Chen H, Meng Q, Huang X. MicroRNA-375 is a therapeutic target for castration-resistant prostate cancer through the PTPN4/STAT3 axis. Exp Mol Med. 2022 Aug;54(8):1290-1305. doi: 10.1038/s12276-022-00837-6.
- Galuppini F, Censi S, Moro M, Carraro S, Sbaraglia M, Iacobone M, Fassan M, Mian C, Pennelli G. MicroRNAs in Medullary Thyroid Carcinoma: A State of the Art Review of the Regulatory Mechanisms and Future Perspectives. Cells. 2021 Apr 20;10(4):955. doi: 10.3390/cells10040955.
- Fan K, Ritter C, Nghiem P, Blom A, Verhaegen ME, Dlugosz A, Ødum N, Woetmann A, Tothill RW, Hicks RJ, Sand M, Schrama D, Schadendorf D, Ugurel S, Becker JC. Circulating Cell-Free miR-375 as Surrogate Marker of Tumor Burden in Merkel Cell Carcinoma. Clin Cancer Res. 2018 Dec 1;24(23):5873-5882. doi: 10.1158/1078-0432.CCR-18-1184.
- Zellinger B, Bodenhofer U, Engländer IA, Kronberger C, Strasser P, Grambozov B, Fastner G, Stana M, Reitsamer R, Sotlar K, Sedlmayer F, Zehentmayr F. Hsa-miR-375/RASD1 Signaling May Predict Local Control in Early Breast Cancer. Genes (Basel). 2020 Nov 26;11(12):1404. doi: 10.3390/genes11121404.
Overexpression of miR-21-5p has been widely implicated in the progression and aggressiveness of multiple cancers. It promotes tumor growth and survival primarily through suppression of PTEN, resulting in PI3K/AKT pathway activation [24], and by targeting PDCD4, which enhances AP-1–mediated transcription of genes driving proliferation and invasion [25]. Additionally, miR-21-5p facilitates epithelial-to-mesenchymal transition via SMAD7 inhibition, and promotes extracellular matrix remodeling through downregulation of RECK, leading to increased MMP2/9 activity [26]. By also modulating apoptosis- and cell cycle-related proteins such as BCL2, TPM1, and SPRY2, miR-21-5p contributes to enhanced metastatic potential and chemoresistance [27].
- Chawra HS, Agarwal M, Mishra A, Chandel SS, Singh RP, Dubey G, Kukreti N, Singh M. MicroRNA-21's role in PTEN suppression and PI3K/AKT activation: Implications for cancer biology. Pathol Res Pract. 2024 Feb;254:155091. doi: 10.1016/j.prp.2024.155091.
- Liu C, Tong Z, Tan J, Xin Z, Wang Z, Tian L. MicroRNA-21-5p targeting PDCD4 suppresses apoptosis via regulating the PI3K/AKT/FOXO1 signaling pathway in tongue squamous cell carcinoma. Exp Ther Med. 2019 Nov;18(5):3543-3551. doi: 10.3892/etm.2019.7970.
- Bautista-Sánchez D, Arriaga-Canon C, Pedroza-Torres A, De La Rosa-Velázquez IA, González-Barrios R, Contreras-Espinosa L, Montiel-Manríquez R, Castro-Hernández C, Fragoso-Ontiveros V, Álvarez-Gómez RM, Herrera LA. The Promising Role of miR-21 as a Cancer Biomarker and Its Importance in RNA-Based Therapeutics. Mol Ther Nucleic Acids. 2020 Jun 5;20:409-420. doi: 10.1016/j.omtn.2020.03.003.
- Bautista-Sánchez D, Arriaga-Canon C, Pedroza-Torres A, De La Rosa-Velázquez IA, González-Barrios R, Contreras-Espinosa L, Montiel-Manríquez R, Castro-Hernández C, Fragoso-Ontiveros V, Álvarez-Gómez RM, Herrera LA. The Promising Role of miR-21 as a Cancer Biomarker and Its Importance in RNA-Based Therapeutics. Mol Ther Nucleic Acids. 2020 Jun 5;20:409-420. doi: 10.1016/j.omtn.2020.03.003. Epub 2020 Mar 13. PMID: 32244168; PMCID: PMC7118281.
Interestingly, hsa-miR-503-5p showed an upward trend in expression with increasing histological severity of dysplasia and progression to laryngeal carcinoma. TargetScan predictions and literature curation identify several high-confidence targets of hsa-miR-503-5p, including the tumor suppressor PDCD4 and key regulators of the cell cycle and apoptosis (CCND1/CCNE1/CDC25A, PUMA, SMAD7, TIMP2) [37, 38]. In our cohort, elevated miR-503-5p expression would be expected to suppress these critical brakes, thereby promoting proliferative expansion, extracellular matrix remodeling and invasion, as well as evasion of apoptosis—hallmarks that parallel the progression from dysplasia to carcinoma. Taken together, these findings support a model in which miR-503-5p upregulation drives disease progression primarily through PDCD4 loss, further reinforced by dysregulation of cell-cycle and survival pathways.
- Shuang Y, Zhou X, Li C, Huang Y, Zhang L. MicroRNA‑503 serves an oncogenic role in laryngeal squamous cell carcinoma via targeting programmed cell death protein 4. Mol Med Rep. 2017 Oct;16(4):5249-5256.
- Caporali A, Meloni M, Völlenkle C, Bonci D, Sala-Newby GB, Addis R, Spinetti G, Losa S, Masson R, Baker AH, Agami R, le Sage C, Condorelli G, Madeddu P, Martelli F, Emanueli C. Deregulation of microRNA-503 contributes to diabetes mellitus-induced impairment of endothelial function and reparative angiogenesis after limb ischemia. Circulation. 2011 Jan 25;123(3):282-91.

Reviewer 2 Report
Comments and Suggestions for Authors
Title
MiRNA profiling in premalignant lesions and early glottic cancer (cancers-3799687).
The authors reported an article for validating and comparing the expression of MiRNA for laryngeal carcinogenesis as primary marker from patient derived samples. The manuscript is written well and will be interesting for scientific community. It can be accepted for publication after minor revisions.
Specific comments
- In the introduction, if the authors can talk about the new techniques for miRNA detection, it would be great.
- Moreover, the miRNA profiling is time-consuming; techniques such as microfluidic, SERS, and Isothermal amplification techniques are available these days. So the profiling is performed by PCR methods by the authors?
- The results contain lots of tables with a lot of data in the tables. It's difficult to go through each. Is there any other representation the authors can make? Since these are important results.
- Is there any way the heat maps can be split and made into bigger versions, so that they will be visible to understandable. Either make two heat maps or arrange them on a single line, with enlarged blocks.
- It seems like there are slight differences in the heat map intensity with all the microRNAs, so how accurately can we separate and identify them?
- Were the identified miRNAs functionally validated to determine their role in the progression of the disease or condition under study?
- What are the predicted target genes of hsa-miR-503-5p, and how might their regulation contribute to the observed increase in histological severity?
Author Response
MiRNA profiling in premalignant lesions and early glottic cancer (cancers-3799687).
The authors reported an article for validating and comparing the expression of MiRNA for laryngeal carcinogenesis as primary marker from patient derived samples. The manuscript is written well and will be interesting for scientific community. It can be accepted for publication after minor revisions.
Specific comments
- In the introduction, if the authors can talk about the new techniques for miRNA detection, it would be great.
We appreciate the reviewer’s suggestion very much and supplemented the introduction section with following paragraph:
Recent advances in miRNA detection technologies have greatly improved the sensitivity, specificity, and speed of profiling these critical regulatory molecules. Traditional methods, such as qRT-PCR and microarrays, remain widely available and relatively affordable, but they can struggle with low-abundance miRNAs and multiplexed analysis. Innovative approaches, including isothermal amplification techniques, CRISPR-Cas-based biosensors, nanotechnology-enhanced platforms like surface-enhanced Raman scattering (SERS) and nanoparticle-assisted hybridization, as well as microfluidic lab-on-a-chip systems, offer enhanced signal amplification, real-time detection, and high-throughput capabilities [13-15]. While some of these cutting-edge platforms provide superior performance, they are often less accessible and more expensive, limiting routine clinical use. Lateral flow assays and electrochemical biosensors offer a balance of cost-effectiveness, rapid detection, and minimal sample requirements, making them promising tools for point-of-care applications. Collectively, these emerging techniques are expanding the potential for accurate, multiplexed miRNA detection in near future, supporting early diagnosis and personalized medicine.
- Dave VP, Ngo TA, Pernestig AK, Tilevik D, Kant K, Nguyen T, Wolff A, Bang DD. MicroRNA amplification and detection technologies: opportunities and challenges for point of care diagnostics. Lab Invest. 2019, 99:452-469.
- Wang, H. A Review of Nanotechnology in microRNA Detection and Drug Delivery. Cells2024, 13, 1277.
- Nafari NB, Zamani M, Mosayyebi B. Recent advances in lateral flow assays for MicroRNA detection. Clin Chim Acta. 2025 1;567:120096.
2. Moreover, the miRNA profiling is time-consuming; techniques such as microfluidic, SERS, and Isothermal amplification techniques are available these days. So the profiling is performed by PCR methods by the authors?
We appreciate the reviewer’s valuable comment. In our study, the miRNA profiling was indeed carried out using PCR-based methods.
3. The results contain lots of tables with a lot of data in the tables. It's difficult to go through each. Is there any other representation the authors can make? Since these are important results.
We thank the reviewer for this valuable suggestion. We are aware that extensive tabular data may make it difficult to quickly grasp the results. Our intention in presenting the data in such detail was to enable comprehensive analysis and to provide a reference point for further comparisons by other researches. The complete tabular data have been retained to ensure the possibility of precise comparative analyses in the future.
4. Is there any way the heat maps can be split and made into bigger versions, so that they will be visible to understandable. Either make two heat maps or arrange them on a single line, with enlarged blocks.
We thank the reviewer for this constructive suggestion. Given the large number of significantly deregulated miRNAs across the histopathological stages and the wide range of 2^(−ΔCt) values (−2.82 to 11.21), we selected a color scale that best represents the overall variability within a single heatmap. This approach allowed us to present all relevant differences in a comprehensive way for this pilot study. We agree that in future studies, once selected miRNAs are validated, enlarged or split heatmaps will be considered to improve clarity.
5. It seems like there are slight differences in the heat map intensity with all the microRNAs, so how accurately can we separate and identify them?
The question was addressed above.
6. Were the identified miRNAs functionally validated to determine their role in the progression of the disease or condition under study?
We thank the reviewer for this important comment. In the present study, the identified miRNAs were not yet functionally validated to directly confirm their role in the progression of laryngeal dysplasia and carcinoma. Our primary aim here was to profile expression patterns and explore their potential associations with histological severity. However, we fully acknowledge the importance of functional validation. We are now planning follow-up experiments using specific miRNA probes and in vitro assays to assess the biological effects of candidate miRNAs on proliferation, apoptosis, and invasion, which will provide deeper mechanistic insight into their contribution to disease progression.
7. What are the predicted target genes of hsa-miR-503-5p, and how might their regulation contribute to the observed increase in histological severity?
We appreciate the reviewer’s suggestion very much and supplemented the discussion section with following paragraph:
Interestingly, hsa-miR-503-5p showed an upward trend in expression with increasing histological severity of dysplasia and progression to laryngeal carcinoma. TargetScan predictions and literature curation identify several high-confidence targets of hsa-miR-503-5p, including the tumor suppressor PDCD4 and key regulators of the cell cycle and apoptosis (CCND1/CCNE1/CDC25A, PUMA, SMAD7, TIMP2) [37, 38]. In our cohort, elevated miR-503-5p expression would be expected to suppress these critical brakes, thereby promoting proliferative expansion, extracellular matrix remodeling and invasion, as well as evasion of apoptosis—hallmarks that parallel the progression from dysplasia to carcinoma. Taken together, these findings support a model in which miR-503-5p upregulation drives disease progression primarily through PDCD4 loss, further reinforced by dysregulation of cell-cycle and survival pathways.

Reviewer 3 Report
Comments and Suggestions for Authors
This study investigated miRNA expression profiles across different stages of laryngeal squamous cell carcinoma progression, from normal tissue to invasive carcinoma. And the authors identified several miRNAs with stage-specific expression patterns that may serve as potential biomarkers. Overall, this manuscript is well-organized. It is recommended that the paper be resubmitted for review after the authors address the following minor points:
The authors emphasized several miRNAs, such as miR-105-5p, miR-516a-5p, miR-212-3p, miR-548d-5p, in the summary, abstract and conclusions. However, in several figures and tables, these miRNAs were not shown,
- Why miRNAs (hsa-miR-105-5p-477865_mir and hsa-miR-516a-5p-478978_mir) identified only in IC and HGD patients were not shown in Table 3 and Figure1?
- Why miRNAs (hsa-miR-212-3p-478318_mir and hsa-miR-548d-5p-480870_mir) identified only in IC, HGD and LGD patients were not shown in Table 4 and Figure3?
Author Response
This study investigated miRNA expression profiles across different stages of laryngeal squamous cell carcinoma progression, from normal tissue to invasive carcinoma. And the authors identified several miRNAs with stage-specific expression patterns that may serve as potential biomarkers. Overall, this manuscript is well-organized. It is recommended that the paper be resubmitted for review after the authors address the following minor points:
The authors emphasized several miRNAs, such as miR-105-5p, miR-516a-5p, miR-212-3p, miR-548d-5p, in the summary, abstract and conclusions. However, in several figures and tables, these miRNAs were not shown,
- Why miRNAs (hsa-miR-105-5p-477865_mir and hsa-miR-516a-5p-478978_mir) identified only in IC and HGD patients were not shown in Table 3 and Figure1?
- Why miRNAs (hsa-miR-212-3p-478318_mir and hsa-miR-548d-5p-480870_mir) identified only in IC, HGD and LGD patients were not shown in Table 4 and Figure3?
We thank the reviewer for pointing this out. The apparent discrepancy arises from the criteria we applied for inclusion of miRNAs in Tables 3–4 and Figures 1–3. These tables and figures were designed to present miRNAs that were consistently expressed across multiple groups and demonstrated significant differential expression patterns in the comparative analyses.
Regarding hsa-miR-105-5p and hsa-miR-516a-5p:
These miRNAs were indeed detected exclusively in the HGD and IC groups and were absent in the control and less advanced lesions. However, the comparison of expression levels between both groups revealed no significant difference, and therefore this types were not included. We decided to point their restricted presence in HGD and IC was highlighted in the Results and emphasized in the Abstract and Conclusions due to their potential relevance as markers of progression.
Regarding hsa-miR-212-3p and hsa-miR-548d-5p:
Similarly, these miRNAs were detected only in IC, HGD, and LGD samples, but absent in the control and ND groups. Table 4 and Figure 3, however, were restricted to miRNAs that showed significant deregulation between the groups being compared in that specific analysis. For this reason, these stage-specific miRNAs were not included in the figures and tables, although we highlighted them in the text and summary due to their potential biological and clinical importance.
In summary, the absence of these miRNAs from the tables and figures reflects the inclusion criteria for the visualizations, not their lack of importance. To improve clarity, we will add a note in the figure legends and Results section to explain this rationale, ensuring readers can readily understand why certain miRNAs emphasized in the text are not displayed in those specific figures or tables.
We supplemented the result section with following paragraph:
The miRNAs highlighted in Table 2 and discussed in the text were not consistently expressed across multiple groups and did not demonstrate statistically significant differences in the specific comparisons presented. Stage- or group-specific miRNAs are mentioned as potentially biologically important biomarkers
